# Dynamic Prediction Model for Initial Apple Damage

**DOI:** 10.3390/foods12203732

**Published:** 2023-10-11

**Authors:** Tao Xu, Yihang Zhu, Xiaomin Zhang, Zheyuan Wu, Xiuqin Rao

**Affiliations:** 1College of Biosystems Engineering and Food Science, Zhejiang University, Hangzhou 310058, China; 12113053@zju.edu.cn (T.X.); 22013052@zju.edu.cn (Y.Z.); 11813014@zju.edu.cn (X.Z.); 3180100311@zju.edu.cn (Z.W.); 2Key Laboratory of On Site Processing Equipment for Agricultural Products, Ministry of Agriculture, Hangzhou 310058, China; 3Key Laboratory of Intelligent Equipment and Robotics for Agriculture of Zhejiang Province, Hangzhou 310058, China

**Keywords:** mechanical damage, cell death zone, damage model, damage parameter, impact energy distribution

## Abstract

Prediction models of damage severity are crucial for the damage expression of fruit. In light of issues such as the mismatch of existing models in actual damage scenarios and the failure of static models to meet research needs, this article proposes a dynamic prediction model for damage severity throughout the entire process of apple damage and studies the relationship between the initial bruise form and impact energy distribution of apple damage. From the experiments, it was found that after impact a “cell death zone” appeared in the internal pulp of the damaged part of *Red Delicious* apples. The reason for the appearance of the cell death zone was that the impact force propagated in the direction of the fruit kernel in the form of stress waves; the continuous action of which continuously compressed the pulp’s cell tissue. When the energy absorbed via elastic deformation reached the limit value, intercellular disadhesion of parenchyma cells at the location of the stress wave peak occurred to form cell rupture. The increase in intercellular space for the parenchyma cells near the rupture site caused a large amount of necrocytosis and, ultimately, formed the cell death zone. The depth of the cell death zone was closely related to the impact energy. The correlation coefficient r between the depth of the cell death zone and the distribution of impact energy was slightly lower at the impact height of 50 mm. As the impact height increased, the correlation coefficient r increased, approaching of value of 1. When the impact height was lower (50 mm), the correlation coefficient r had a large distribution range (from 0.421 to 0.983). As the impact height increased, the distribution range significantly decreased. The width of the cell death zone had a poor correlation with the pressure distribution on the impact surface of the apples that was not related to the impact height. In this article, the corresponding relationship between the form and impact energy distribution of the internal damaged tissues in the initial damage of *Red Delicious* apples was analyzed. This analysis aimed to provide a research concept and theoretical basis for more reliable research on the morphological changes in the damaged tissues of apples in the future, further improving the prediction accuracy of damage severity.

## 1. Introduction

Mechanical damage from falls, collisions, squeezing, vibrations, and so on is one of the main causes of fruit loss in fruit picking, transport, processing, and other parts of the process [1]. Especially in the early stages of damage, due to the damaged parts being difficult to detect, the damaged parts further rot and deteriorate and spread to surrounding fruits during storage and transportation. As a result, serious losses will occur that affect producers and consumers. Therefore, a large number of studies have been conducted on damage simulation to provide a theoretical basis for damage prevention and predictions of damage severity.

Damage severity is an important indicator of fruit grading. Usually, the damage severity is quantitatively measured via damage area and damage volume. The damage area is easily measured. Therefore, the damage area is used as an indicator for fruit grading in China [2], the United States [3], and some European countries [4]. However, it cannot fully characterize damage severity, so damage volume containing damage surface information and internal depth information is the most common indicator of damage severity in cutting-edge research on damage prevention and prediction [5].

In several studies, the volume of fruit damage was predicted by the impact model of the ellipsoid hypothesis [6,7]. In the literature, objects to be predicted have included apples that impacted an aluminum plate for 36 h [8], apples that impacted cardboard for 48 h [9], olives that impacted a steel plate for 2 h [10], apples that impacted a polyethylene foam plane for 4 days [11], pomegranates that impacted a wooden board for 24 h [7], and apples that were impacted by a pendulum made of wooden balls for 24 h [12]. The damage prediction of the above objects was based on the evaluation of the damage severity of the damaged part of the fruit for a period of time after the fruit was impacted by a planar or spherical impact surface. Compared with a damage simulation scenario in a laboratory, the impacted surfaces of various forms occur in an actual damage scenario, with a complex distribution of impact surface force meaning that the bruise that forms inside an apple changes accordingly. As a result, a single model cannot be fully matched and applied. At the same time, the bruise that forms inside the apple dynamically changes and the internal damage severity varies with the different times when damage occurs. Especially in the initial damage, the changes are more intense. Using static models to express damage severity fails to meet current research needs. As a consequence, there is a need for carrying out research on the form of apple damage and the expansion process of damaged tissue to lay the foundation for a dynamic model of damage volume throughout the entire process of damage. Examining the relationship between the initial bruise form of apples and the distribution of impact energy is the basis for the study of bruise form and the expansion process of damaged tissue.

In this study, the corresponding relationship between the initial bruise form of the internal tissue and the impact energy distribution was analyzed based on *Red Delicious* apples as the study object, and the relationship between the damage parameter and the damage force was established to provide a research concept and theoretical basis for more reliable follow-up studies on the morphological changes of the damaged tissue and further improvement of the prediction accuracy of damage severity.

## 2. Materials and Methods

### 2.1. Research Materials and Instruments

A batch of *Red Delicious* apples was purchased from the local wholesale fruit market for the experimental material. All the selected apples were similar in size, weight, hardness, and ripeness [13] and without cracks, rot, heterocarpy, or other damage. A total of 140 apples were randomly selected (used for the mechanical damage experiment; apples needed for other experiments were counted separately), and the diameter, height, and weight of the apples were measured one by one. By measuring the conical shape of the *Red Delicious* apples (fruit shape indexes were within 0.9–1.0), the diameters of the upper part, middle part, and lower part in the direction of the axle wire were obtained (Figure 1). The dimensions of each apple were measured by taking extreme values. The measurements for the upper and lower parts were taken at the maximum diameter, whereas the measurement for the middle part was taken at the minimum diameter. Furthermore, because of the obvious protrusion of the five ridges, the apples were measured once per 120° rotation for a total of 9 times. As for the measurement of the heights, the apples were measured once every 120° rotation and the average value was taken. The average diameters of the upper part, middle part, and lower part of the apples were 74.08 mm, 64.51 mm, and 64.09 mm (standard deviations were 2.46 mm, 2.23 mm, and 2.71 mm), respectively, the average height was 88.54 mm (standard deviation was 3.38 mm), and the mean weight was 233.94 g (standard deviation 9.76 g) [13,14,15]. To avoid potential damage to the apples during the hardness measurement process and its impact on the experimental results, an additional set of 50 apples was selected specifically for apple hardness testing. For each apple, one measurement point was chosen at the upper, middle, and lower parts and rotated by 90 degrees each time. Hardness measurements were performed using a GY-4 fruit hardness tester (Zhejiang TopCloud Agricultural Technology Co., Ltd., Hangzhou, China) with an 11.1 mm probe. The average hardnesses of the upper part, middle part, and lower part of the apples were 14.86 N, 15.77 N, and 15.33 N (standard deviations were 1.76 N, 2.55 N, and 1.93 N), respectively.

The test was conducted using the test apparatus shown in Figure 2. On the basis of the MY-A721 impact test apparatus (Dongguan Mingyu Intelligent Technology Co., Ltd., Dongguan, China), an array pressure distribution detection system (Shenzhen Ligan Technology Co., Ltd., Shenzhen, China) and customized plastic fruit holders were used. The MY-A721 impact test apparatus comprised a base, a ruler, a guide rod, a buckle, and an impact hammer. The impact hammer had a diameter of 80 mm, which was more than four times the size of the apple contact dimension during the impact process, and a weight of 500 g; there was an RPPS-1600 sensor fixed on its lower surface. This sensor was an array pressure sensor [16], with a sensor array distributed with 1600 sensing points (40 × 40). There was one sensing point every 1.9 mm × 1.9 mm, and the resistance value of each sensing point decreased with the increase in pressure acting on it. The pressure distribution information could be captured at a frame rate of 60 fps by using an RPPS-1600 sensor with a collector and corresponding software. Before using the array pressure sensor, a calibration block provided by the manufacturer was used (80 mm × 80 mm × 10 mm) for the collection of the calibration data. This was performed to reduce the data collection error of the sensor.

In the test, an OLYMPUS CKX53 microscope (Olympus Corporation, Tokyo, Japan) with a color microscopic digital imaging system of 19.8 million pixels (model D198, Hangzhou Biogroup Technology Co., Ltd., Hangzhou, China) was employed to obtain microscopic images. The imaging system’s CellView X64 imaging software (Hangzhou Biogroup Technology Co., Ltd., Hangzhou, China) was used to stitch the microscopic images. The image resolution was 5472 pixels × 3648 pixels, and images were saved in TIFF format.

### 2.2. Observation and Localization Methods of Damage

Due to the thick skin and dark color of *Red Delicious* apples (bright red or thick red) (Figure 3a), it was hard to observe and localize the damaged part. However, an aqueous solution of vitamin B2 could produce a distinct fluorescence in the ultraviolet band, did not penetrate into the interior of the apples [17], and did not affect the follow-up experiments. Therefore, a 10% aqueous solution of vitamin B2 was applied to the impact surface of the sensor. Upon impact, the aqueous solution of vitamin B2 adhered to the surfaces of the apples. The damaged location could be observed under an ultraviolet lamp (395 nm) (Figure 3b).

### 2.3. Experiment of Mechanical Damage

As shown in Figure 2, the upper part of the apple was selected as the impact region after the sample was stabilized in a fruit holder. Then, a 10% aqueous solution of vitamin B2 was applied to the lower surface of the array pressure sensor. The ruler slider of the impact test apparatus was moved to the specified height (50 mm, 75 mm, 100 mm, 125 mm, 150 mm, 175 mm, or 200 mm). The impact hammer was released to impact the apple inside the fruit holder, causing damage. At the same time, the dynamic information collection of the pressure distribution was completed with the array pressure distribution detection system during the damage process. When the impact height was below 50 mm, the apples remained undamaged. However, when the impact height exceeded 200 mm, the damaged area of the apples exhibited ruptured skin and significant deformation that could not be restored. Therefore, seven heights ranging from 50 mm to 200 mm with an increment of 25 mm were selected for mechanical damage experiments. The preparation of 20 damaged apple samples and dynamic information collection of pressure distribution were accomplished for each height.

### 2.4. Collection of the Microscopic Images of the Damaged Parts

The damaged parts were evaluated under an ultraviolet lamp (395 nm) within 10 s of the apples being impacted. The central point of the impact position and the plane located at the axle wire of the apple were used as the segmentation plane for apple segmentation (Figure 4). The sections of the damaged parts were regarded as objects to make tissue sections. The sections were cleaned with phosphate-buffered saline (PBS) and then stained with 0.04% trypan blue solution at room temperature for about 30 s. After dyeing, the sections were washed with PBS again [18]. The microscope was used to collect a microscopic image of the damaged parts of the apples, and the CellView imaging software was used to stitch the images of the tissue sections. In order to avoid the development of a bruise form in damaged parts of the apples, the entire process of collecting damage images was completed within 90 s.

### 2.5. Determination of Cell Death Zone

#### 2.5.1. Extraction of Damage Area

The cell death zone is composed of concentrated dead cells, and cell death is also caused by the process of making tissue sections. All the dead cells appear blue in the microscopic image (Figure 5a). The boundary between the cell death zone and other parts was unclear because the dead cells that resulted from the process of making the tissue sections covered the entire surface of the tissue sections. The image analysis found that color saturation could be used as an indicator to delineate the boundary of the cell death zone. As a result, after the HSV color space channel was separated, the S (saturation) channel was used to extract the cell death zone (Figure 5c).

The S-channel component (Figure 5c) was subjected to threshold segmentation, resulting in a segmented image (Figure 6a). The segmented image was further processed using morphological operations of opening and closing (Figure 6b), extraction of the largest connected component, and hole filling to achieve precise segmentation of the damage area (Figure 6c).

#### 2.5.2. Collection of Damage Data

The 2D coordinates of the peel surface in the S-channel component (Figure 5c) and the 2D coordinates of the edge of the cell death zone (Figure 6c) were collected using Matlab R2022a (MathWorks, Inc., Natick, MA, USA) to calculate data such as the distance from the cell death zone to the peel surface and the width of the cell death zone (hereinafter referred to as the depth and the width of the cell death zone). However, when taking data in the Y-axis direction, there may be discontinuities in the cell death zone. The midpoint of the two furthest points was chosen as the central position of the cell death zone for the depth calculation, and the sum of the widths of all the cell death zones in this direction was chosen as the width. Taking the situation in Figure 7 as an example, (Y_1_ + Y_4_)/2 was selected as the central position of the cell death zone for the depth calculation, and (Y_2_ − Y_1_) + (Y_4_ − Y_3_) was selected as the width.

### 2.6. Collection of the Data for Impact Energy Distribution

The calibration data were used to calibrate the pressure distribution information collected by the array pressure sensor.

Equations (1)–(3) are derived from Newton’s second law and the kinetic energy theorem. They are used to convert the pressure distribution data of the test sample into impact energy distribution data. The specific process is as follows:

(1) To collect the data of pressure distribution for the effective frames;

(2) To calculate the movement speed *V_1_* of the impact hammer in the first frame. The velocity of the impact hammer impacting the apple in the form of free fall is approximately considered as the average velocity of the impact hammer in the first frame of the effective frames;

(3) To calculate the movement speed *V_f_* and movement distance *S_f_* of the impact hammer for each frame. The velocity of the impact hammer varies nonlinearly, and the movement speed and movement distance of the impact hammer are calculated for each frame based on the pressure distribution data of the effective frames;

(4) To calculate the impact energy *E*(*i,j*) absorbed by the corresponding apple positions at various sensing points. The impact energies absorbed by the corresponding apple positions at various sensing points are calculated through the pressure distribution data and movement distance of the effective frames.
(1)$$V_{f}=\left\{\begin{array}{cc} \sqrt{2gh} & ,\;f=1\\ \sqrt{2gh}-\frac{\sum_{x=1}^{f-1}\sum_{i=1}^{40}\sum_{j=1}^{40}F(i,j,x)}{m}\cdot \frac{1}{Z} & ,\;f>1 \end{array}\right.$$
(2)$$S_{f}=V_{f}\cdot \frac{1}{Z}$$
(3)$$E(i,j)=\sum_{x=1}^{l}\left(F(i,j,x)\times S_{x}\right)$$
where *V_f_* refers to the velocity of the impact hammer at frame *f*, m/s; g refers to the gravitational acceleration, 9.8 m s^−2^; *h* refers to the impact height of the impact hammer, m; *F*(*i*,*j*,*f*) denotes the pressure data of the *f*-th frame at position (*i,j*) collected by the array pressure sensor; *i*,*j* denotes the horizontal and vertical coordinates of the sensing point of the array pressure sensor; *f* represents the number of frames; m represents the weight of the impact hammer; Z refers to the frame rate, 60 fps; *S_f_* refers to the distance traveled by the impact hammer at frame *f*, m; *E*(*i*,*j*) denotes the impact energy at position (*i,j*), J; and *l* represents the number of effective frames.

A 3D diagram of the impact energy distribution was drawn using Matlab (Figure 8a), and the data of the impact energy distribution of the location of the impact central point parallel to the axis of the apple was used to form the curve of impact energy distribution (Figure 8b).

The correlation analysis was conducted on data such as the depth, width, and the impact energy distribution of the cell death zone obtained in Section 2.5.

### 2.7. Experiment of Cell Size Distribution

The fabrication of the tissue sections and collection of the microscopic images of the undamaged parts of 10 apple samples were completed in the same way as described in Section 2.4. CellView software was used to measure the cell size distribution of the microscopic images. Because the fact that cells with a depth of more than 1.2 mm tended to have a spherical shape and similar sizes (without significant difference), a depth of 0–1.8 mm was selected for measuring the cell size distribution. A total of 18 measurement depths were selected uniformly within the range of the target depths for the measurement of the cell sizes. The measurement data included the depth, length of the major diameter, and length of the minor diameter for the central position of cells, and the average diameter was calculated.

### 2.8. Experiment for the Determination of Biological Yield Points

In this experiment, the TA.XT plus C texture analyzer (SMS Engineering (China) Ltd., Shanghai, China) and the supporting P35 flat probe were used for the determination of the biological yield points. Additionally, based on the compression test standard in ASAE S368.4 DEC2000 (R2008) [19], the pre-compression speed was set to 5 mm min^−1^, the downward compression speed to 1 mm min^−1^, the upward speed after compression to 1 mm min-1, the minimum perceived force to 0.05 N, and the loading displacement to 5 mm. The apple sample was cylindrical in shape, with a diameter of 15 mm and a length of 15 mm. It was placed upright for compression testing and force–time data measurement was completed. The experiment was repeated for 10 times, the external force data of the apples were collected through the characteristic curve when the biological yield points were met and the average value was calculated.

## 3. Results and Discussion

### 3.1. Theoretical Analysis of the Mechanism of the Phenomenon of the Cell Death Zone

The experiment showed that there was a tissue rupture surface at a certain depth under the skin of the damaged parts of *Red Delicious* apples. The 3D tissue rupture surface presented a zoned feature on the tissue sections (cell death zone). This was similar to the phenomenon discovered by Mitsuhashi Gonzalez et al. in a study of the relationship between apple damage contours and tissue type and structure [20]. The damage occurred at a certain depth below the skin and developed in both directions of the apple skin and core, forming a relatively stabilized damage area.

The phenomenon of the cell death zone was analyzed from the perspective of cell and tissue structure. After the mechanical damage occurred, the cells and tissue structure at the impacted site were damaged, but the internal damage tissue form varied among different varieties of apples. Li et al. found that the rupture surfaces of the pulp of Fuji and Ruiyang fruits were governed by cell rupture, whereas the rupture surfaces of the pulp of Qinguan fruits were governed by intercellular disadhesion [21]. Based on this, they speculated that the rupture surfaces for the varieties with crispy pulp were governed by cell rupture, whereas the rupture surfaces for the varieties with loose pulp were governed by intercellular disadhesion. The main reasons for intercellular disadhesion were as follows: (1) the pulp had a strong cell wall or weak intercellular connections [22] and (2) the pulp cells were larger. The larger the cell, the larger the intercellular space and the smaller the adhesion between adjacent cells [23]. The intercellular space was the weakest point in the tissue, and the degree of cell rupture near the space was exacerbated greatly due to the increase in intercellular space caused by intercellular disadhesion [20].

The impact process of *Red Delicious* apples was analyzed from the perspective of stress waves. The impact force propagated in a wave form in the direction from the peel to the core [24]. As apple damage involved both the elastic and plastic stages [25], during the impact process, when the stress was linearly related to strain, elastic waves propagated in the medium [24]. When the stress was non-linearly related to strain, plastic waves propagated in the medium [24]. During the process of constructing the apple damage model, the impact process was simplified into a completely elastic impact stage without damage and a plastic impact stage that caused damage [26]. The stress wave was simplified as a one-dimensional longitudinal wave and propagated unidirectionally vertical to the impact plane. During the impact process, when the stress wave passed through the peel and propagated towards the core in the pulp, a completely elastic impact process occurred first. During the elastic impact process, the amplitude of the stress wave attenuated slowly and the velocity of the stress wave decreased. As it transformed into plastic deformation, the amplitude of the stress waves attenuated rapidly [27].

From the perspective of energy, the impact process of *Red Delicious* apples could be roughly divided into three stages [28,29]. (1) Stage I: Under the action of stress waves, the cells of the apple pulp were compressed and were in the elastic impact stage. During this stage, the energy absorbed was mainly stored in the interior of the apple pulp in the manner of elastic resilience [30]. (2) Stage II: Under the continuous action of the stress waves, the compressive strength of the apple pulp was lower than the strength of the stress waves, resulting in damage evolution, accumulation, and the expansion of the originally intercellular space [30]. (3) Stage III: Under the continuous action of the stress waves, the energy absorbed by elastic deformation reached the limit value, the parenchyma cells at the location of the stress wave peak ruptured, and the stored elastic energy was released.

Based on the above literature analysis and experimental results, it could be determined that the cell death zone was caused when impact occurred and the impact force propagated in the form of stress waves from the peel to the core. Under the action of the stress waves, the apples underwent elastic deformation and the cell tissue of the pulp was compressed continuously. The amplitude of the stress waves attenuated slowly, the velocity of the stress waves decreased slowly, and the impact energy transformed into elastic resilience gradually. Under the continuous action of the stress waves, the increase in the disturbance area generated by the propagation of the stress waves failed to meet the conversion requirements of impact energy and elastic energy. Therefore, the energy absorbed by elastic deformation in the area disturbed by the stress waves reached the limit value [31]. The parenchyma cells at the location of the stress wave peak first experienced intercellular disadhesion, resulting in rupture. The parenchyma cells near the rupture site had a significantly increased rupture rate due to an increase in intercellular space, and a cell death zone was ultimately formed. For the occurrence of a cell death zone, the following conditions need to be met: (1) the apple pulp cells need to be characterized by large cells, thin cell walls, and weak intercellular connection; (2) the peak value of the impact force was greater than the yield force; and (3) the impact energy was greater than the energy absorbed by the elastic deformation of the apple pulp.

In addition, the cell death zone was formed by concentrated cell death. In the initial impact damage of apples, the causes of internal cell death could be divided into two types: (1) direct impact resulting in cell rupture and necrocytosis [20] and (2) impact causing intercellular disadhesion, loss of the protection of the cells on both sides of the rupture zone, and necrocytosis [20,21].

### 3.2. Relationship between Damage Parameters and Impact Energy Distribution

Figure 9 shows the distribution of cell sizes at different depths below the apple skin. The depth ranging from 0 to 1.2 mm corresponded to the apple peel, whereas the depths greater than 1.2 mm corresponded to the apple pulp. The apple skin was composed of a wax layer, cuticle membrane, keratinized epidermal cells, and closely packed sclerenchyma (about six layers of sclerenchyma cells) [17], and the pulp was mainly composed of parenchyma cells [17,23]. In addition to differences in the thicknesses of cell walls, there were also significant differences in size between sclerenchymatous cells and parenchyma cells. Compared with sclerenchymatous cells, parenchyma cells were characterized by being large and having thin cell walls and weak intercellular connections. Therefore, the pulp part was more prone to becoming part of the cell death zone. As shown in Figure 10, the minimum depth of the cell death zone measured in the experiment was 1.23 mm, indicating that the cell death zone mainly occurred in apple pulp. During the impact process, the impact force was transmitted to the parenchyma cells through the fruit peel [32] and the intercellular disadhesion between the parenchyma cells formed a cell death zone.

The depth of the cell death zone refers to the distance from the intercellular disadhesion site between parenchyma cells to the fruit peel. As shown in Figure 10, when the impact height was 125 mm, the depth distribution of the cell death zone has a good correlation with the impact energy distribution at the corresponding impact location of the apple (the correlation coefficient r is 0.913). During the propagation of stress waves within an apple, the impact energy is converted into elastic resilience in the disturbed area and remains unchanged in the undisturbed area [24]. Under the continuous action of the stress waves, the disturbed area within the apple increased rapidly, the amplitude of the stress waves attenuated slowly, and the velocity of the stress waves decreased gradually until the energy absorbed by elastic deformation in the disturbed area reached the limit value and plastic deformation occurred. Therefore, there was a certain correlation between the impact energy and the disturbance distance of the stress waves, i.e., the depth of the cell death zone.

The cell death zone is formed by the mass death of parenchyma cells near the rupture site [20,21]. The cell death zone develops from the center of the rupture to both sides, and its width towards both sides is regarded as the width of the cell death zone. As shown in Figure 11, the width of the cell death zone was basically independent of the impact energy on the impact surface of the apples; the correlation coefficient r was only 0.213.

As shown in Figure 12, the depth of the cell death zone has a good correlation with the impact energy distribution on the impact surface of the apples. At an impact height of 50 mm, the correlation coefficient between the depth of the cell death zone and the impact energy distribution was slightly lower. An increase in the impact height would result in an increase in the correlation coefficient between the depth of the cell death zone and the impact energy distribution, and its correlation coefficient r would approach a value of 1. During the process of impact damage in the apples, the impact force and energy at the edge of the impact were weaker than those at the central position. According to data such as the impact force distribution and depth of the cell death zone calculated based on the data collected from sensor sensing points, it could be seen that the impact force values of some sensing points were less than the yield force (the external force measured through experiments when the apple reached the biological yield point was about 8.75 N) and the depth of the corresponding site was zero. Consequently, the depth of the cell death zone had a poor correlation with the impact energy distribution. As the impact height decreased, so did the damage area, and the correlation coefficient between the impact height and damage area further decreased. When the impact height was small (50 mm), the distribution range of the correlation coefficient r was large (from 0.421 to 0.983). An increase in the impact height would result in a significant decrease in the distribution range. When the impact height was small, the damage severity was greatly influenced by the differences in individual apples, such as the maturity, impact surface curvature, and distribution of intercellular spaces in the pulp [33]. Therefore, the smaller the impact height, the larger the distribution range of correlation coefficients.

As shown in Figure 13, the width of the cell death zone has a poor correlation with the impact energy distribution on the apple impact surface. The correlation coefficient r of all the samples in the experiment was less than 0.479 and independent of the impact height. The cell death zone is formed by the mass death of parenchyma cells near the rupture site [20,21]. After the occurrence of cell rupture, elastic energy is rapidly released; this release process has minimal impact on thin-walled cells. The appearance of a rupture is the main cause of extensive cell death in thin-walled cells. The thin-walled cells near the rupture site lose protection due to an increase in intercellular gaps, resulting in cell rupture and necrosis. The magnitude of the rupture rate is related to intrinsic factors such as apple variety and ripeness and is essentially unrelated to the impact height, which refers to the magnitude of elastic energy.

## 4. Conclusions

In this study, it was found that after impact there was surface tissue rupture with concentrated dead cells in the pulp of the damaged part of *Red Delicious* apples. The surface of the 3D tissue rupture was presented as a strip at the angle of the tissue section, and this strip was called the cell death zone. Based on this, a theoretical analysis was conducted on the mechanism of the cell-death-zone phenomenon and a new damage model was proposed. The reason for the appearance of the cell death zone was that the impact force propagated in the direction of the fruit kernel in the form of stress waves, under whose continuous action the pulp’s cell tissue was continuously compressed. When the energy absorbed by elastic deformation reached the limit value, the intercellular disadhesion of parenchyma cells at the location of the stress wave peak occurred, thereby causing cell rupture. The increase in the intercellular space for the parenchyma cells near the rupture site caused a large amount of necrocytosis and, ultimately, formed the cell death zone.

The relationship between the damage parameters (the depth and width of the cell death zone) and impact energy distribution of the internal damaged tissues after impact damage was explored based on the phenomenon of the cell death zone and the damage model. It was found that the depth of the cell death zone was closely related to the impact energy. The correlation coefficient r between the depth of the cell death zone and the distribution of impact energy was slightly lower at the impact height of 50 mm. As the impact height increased, the correlation coefficient r increased and approached a value of 1. When the impact height was lower (50 mm), the correlation coefficient r had a large distribution range (from 0.421 to 0.983). As the impact height increased, the distribution range significantly decreased. The width of the cell death zone had a poor correlation with the pressure distribution on the impact surfaces of the apples, which was not related to the impact height.

## Figures and Tables

**Figure 1 foods-12-03732-f001:**
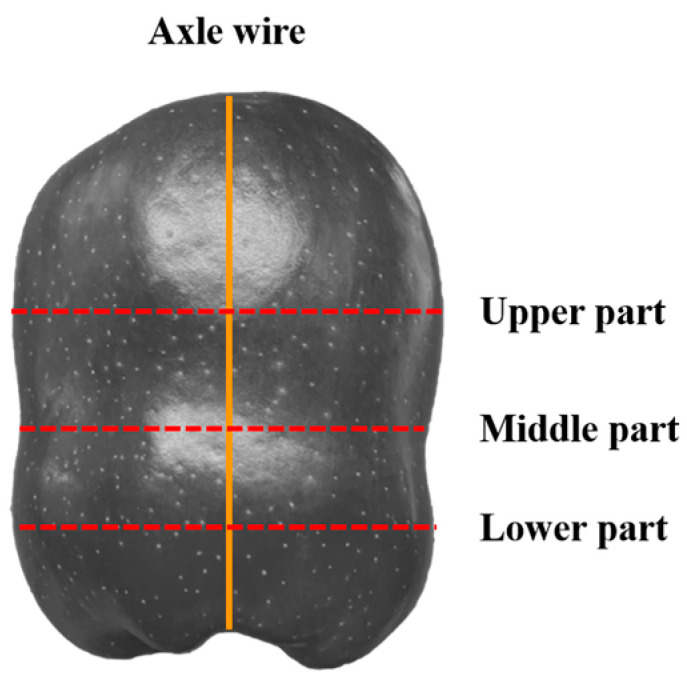
Distribution of apple diameter measurement positions.

**Figure 2 foods-12-03732-f002:**
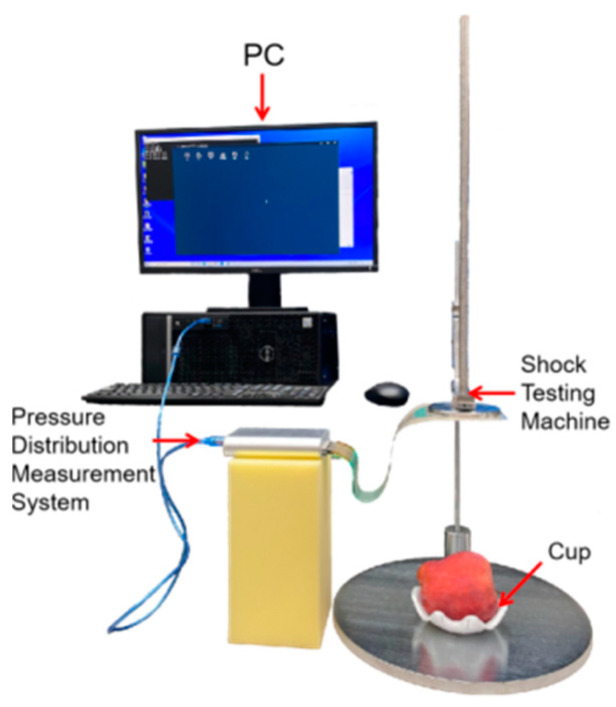
Impact test apparatus.

**Figure 3 foods-12-03732-f003:**
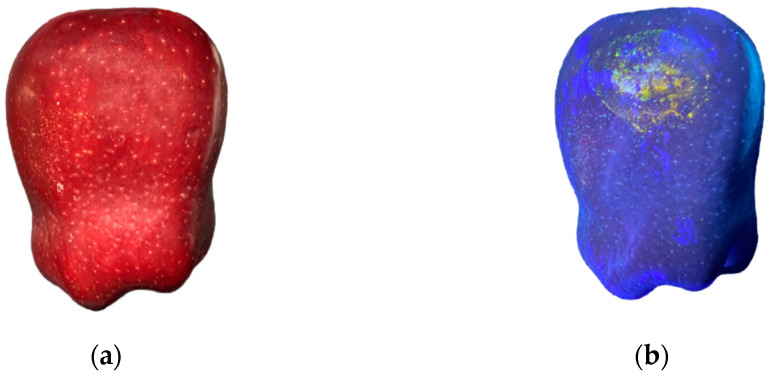
The damaged parts of a *Red Delicious* apple under different lights: (**a**) sunlight; (**b**) ultraviolet.

**Figure 4 foods-12-03732-f004:**
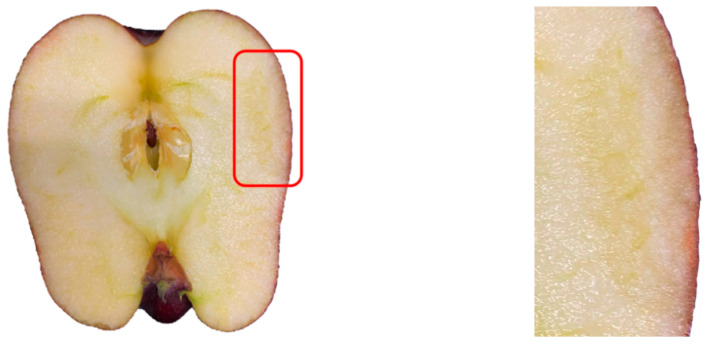
The injured part sectional drawing of an apple.

**Figure 5 foods-12-03732-f005:**
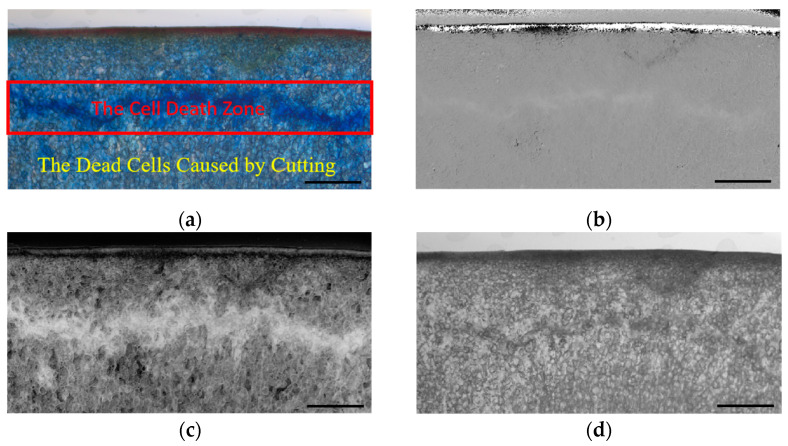
Three-channel component map of HSV color space: (**a**) original image; (**b**) H-channel component; (**c**) S-channel component; (**d**) V-channel component. Scale bar: 2 mm.

**Figure 6 foods-12-03732-f006:**
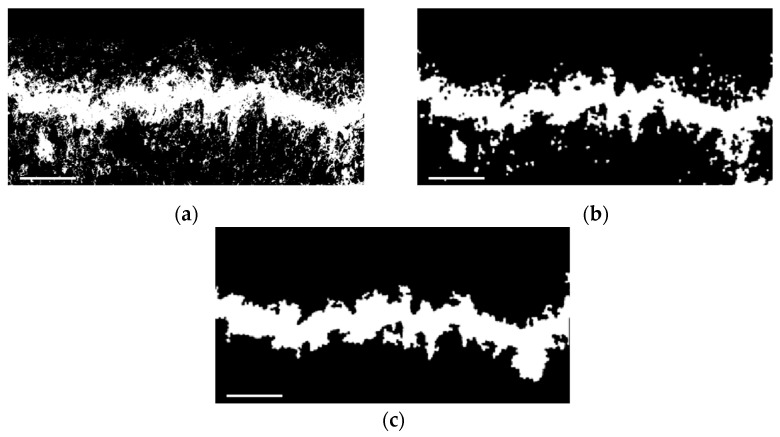
The process of feature extraction for the cell death zone: (**a**) binary image; (**b**) binary image after performing the opening and closing operations; (**c**) the accurately segmented image of the damage area. Scale bar: 2 mm.

**Figure 7 foods-12-03732-f007:**
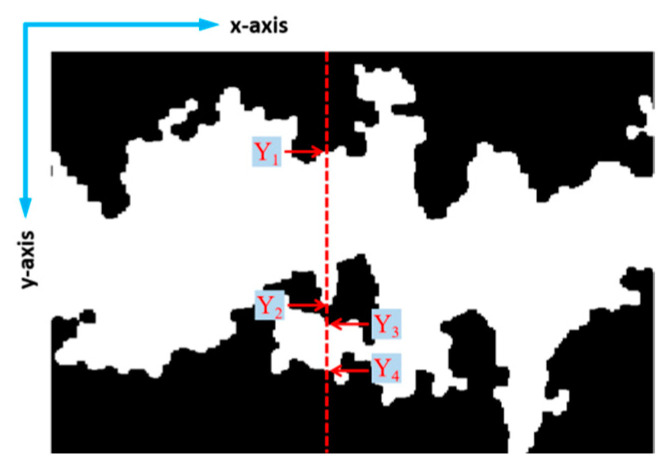
The 2D coordinates of the cell death zone.

**Figure 8 foods-12-03732-f008:**
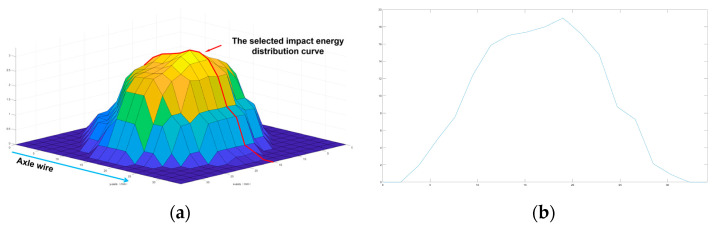
The impact energy distribution and slices of the damaged parts of *Red Delicious* apples: (**a**) 3D diagram of impact energy distribution; (**b**) curve of impact energy distribution.

**Figure 9 foods-12-03732-f009:**
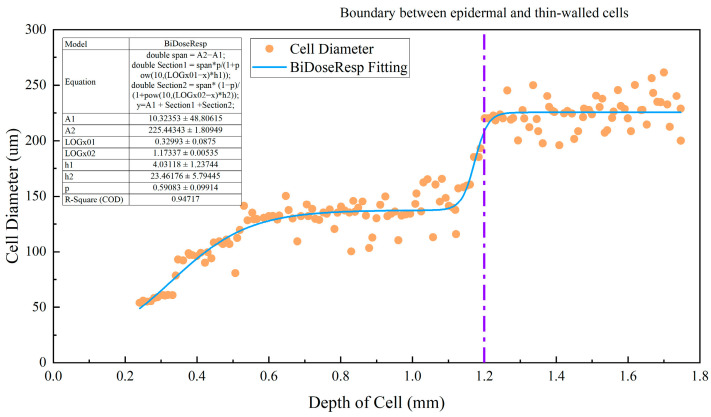
Cell diameter distribution at different depths. The purple line indicates the boundary between epidermal and thin-walled cells.

**Figure 10 foods-12-03732-f010:**
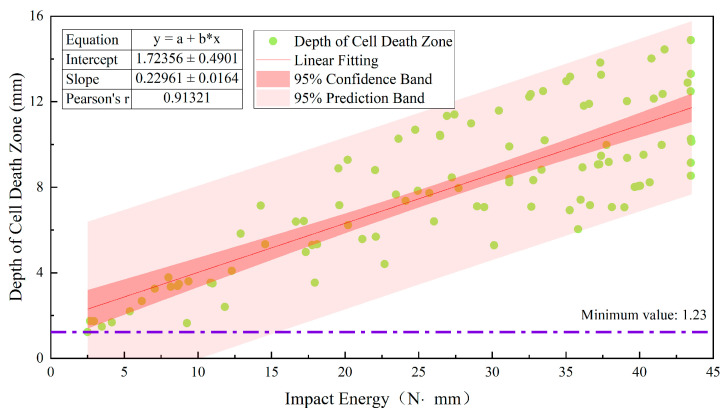
Relationship between impact energy and the depth of the cell death zone. The purple line indicates the minimum depth of the cell death zone measured in the experiment.

**Figure 11 foods-12-03732-f011:**
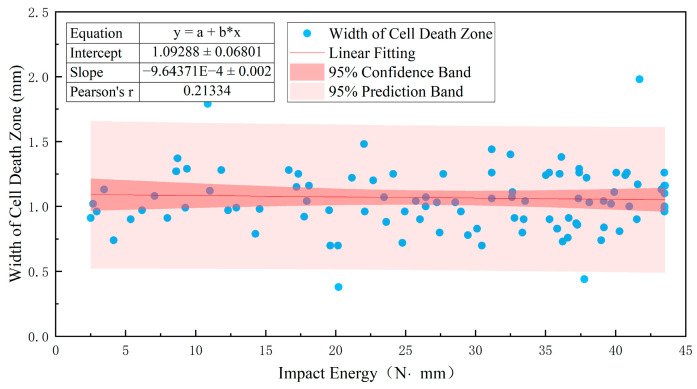
Relationship between impact energy and width of cell death zone.

**Figure 12 foods-12-03732-f012:**
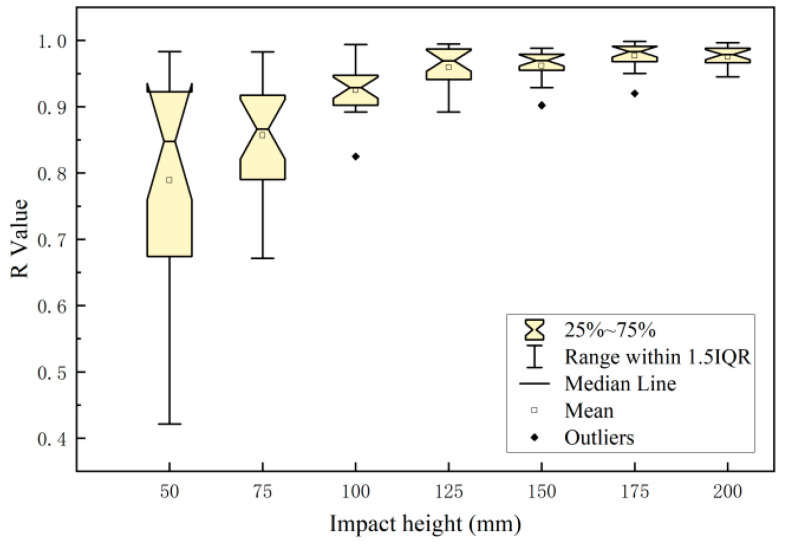
Correlation between the depth of the cell death zone and pressure distribution at different impact heights.

**Figure 13 foods-12-03732-f013:**
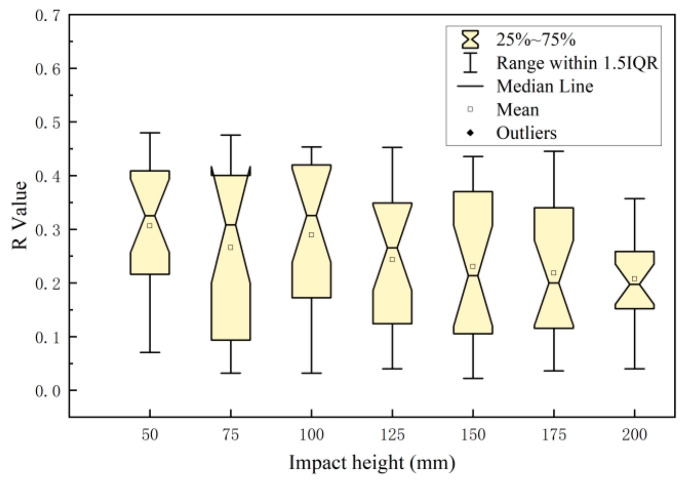
Correlation between the width of the cell death zone and pressure distribution at different impact heights.

## Data Availability

The data used to support the findings of this study can be made available by the corresponding author upon request.

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
