# Peer review of "Dynamic Prediction Model for Initial Apple Damage"

_foods, 2023, doi:10.3390/foods12203732_

Round 1

Reviewer 1 Report

please kindly revise followed my comment in attachment

Author Response

The attachments are divided into three parts: Author’s Reply, Revised Manuscript, and Manuscript with Tracked Changes. Please review and provide your comments and suggestions. Thanks!

Reviewer 2 Report

Please consider the following list:

- For the sample selection, how does the author ensure that the samples are of the same maturity? The different hardness of the sample may affect the impact and bruise susceptibility. 

- The author provided information on the size and weight of the samples; it would be better to provide some information about hardness and ripeness, as mentioned in line 87.

- Some information on the experimental apparatus/system used in the study is missing (lines 103-104).

- The author should provide some reference on the use of the aqueous solution of vitamin B2 for locating impact bruise damage (section 2.2).

- The figure 3 caption: “(a) Sunshine” should read "(a) Sunlight”.

- The author should explain why the specific heights of the impact test ranged between 50-200 mm (section 2.3).

- Details of the CellView software should be mentioned in Section 2.4.

- In Section 2.5: why did the author select only the HSV color space for determining the damage area? Is it possible to use a different color space, e.g., RGB, HSI, or Lab?

- The details on the image, e.g., file type, size, and dpi, should be clarified.

- What’s the criteria (threshold value, thresholding algorithm) for converting the S-channel image to a binary image?

- The version of Matlab software should be mentioned (line 175).

- The binary image of the tissue section of the damaged part should be included in Figure 8 to make it easier to see the distribution clearly.

- The details of the texture analyzer should be mentioned (line 225).

- The author mentions a low correlation between the width of the cell death zone and impact, but with no discussion, it would be better to explain more or find the literature to support the findings.

Author Response

(The authors gave the same response as above.)

Reviewer 3 Report

The manuscript (foods-2644680) aims to develop a dynamic model for predicting apple damage. Current models do not adequately represent actual damage scenarios. Experiments conducted on 'Red Delicious' apples unveiled a 'cell death zone' in the pulp postimpact. This zone emerges when stress waves from the impact lead to cell rupture. The depth of this zone correlates with impact energy, with the correlation coefficient approaching 1 at elevated impact heights. These findings provide valuable insights for upcoming research on apple tissue damage and the enhancement of damage prediction accuracy.

The sections of the manuscript, including the introduction, materials and methods, results and discussion (treated as a single topic), and conclusions, are generally verbose and unclear. The authors do have merit in their work on predicting damage information. However, the models are not aptly suitable, given the generated graphs. Some models are also quite peculiar, attempting to fit the data (e.g., Figure 11).

The quality of the English used is poor. I often found myself lost and unable to understand the authors’ explanations. I strongly suggest reviewing the “Instruction for Authors” and adhering to the standardization in both the manuscript and references, e.g., use “Figure” not “Fig.” Consider improving the flow of ideas and avoid excessive repetition of the word “damage.” Reorganize sentences and ideas for clarity and conciseness.

L12: Redundant use of “damage”

L56-61: What is the context of aluminum?

Figure 5a-d needs a scale bar added. In fact, all images depicting damage or analyses require scale bars and clear descriptions in the figure captions.

What contributions do graphs 9 and 10 make to the work?

The quality of the images is not suitable, especially Figure 11. There is an overload of information presented in the form of an inset. What does the blue line represent in terms of data distribution, and what adjustments were made? How did the authors perform the necessary adjustments?

Figures 14 and 15: I suggest the authors add a statistical test to compare the r-values vs impact height. I did not quite grasp the purpose. Why was this applied in the study?

Other questions:

What advancements have the authors made concerning the initial dynamics for damage prediction models in apples?

Why did the authors only work with the “Red Delicious” variety for prediction, and why were other data (chemical, biological, besides other physical damages) not considered?

What do the authors mean by impact energy and “cell death zone”?

In Figure 3, what do the authors imply with this false colouring?

The English requires extensive correction.

Author Response

(The authors gave the same response as above.)

Round 2

Reviewer 3 Report

I appreciate the authors' responses. The changes were significant; however, small changes are still not necessary. For example, the captions of the figures should be better rewritten to reflect what the figure represents.

Minor changes for grammar.